# OpenReview forum: "LMSYS-Chat-1M: A Large-Scale Real-World LLM Conversation Dataset"
_ICLR.cc/2024/Conference — ICLR 2024 spotlight_

### Official Review · Reviewer_M7pJ · 2023-10-29

**Soundness:** 2 fair
**Presentation:** 3 good
**Contribution:** 4 excellent
**Rating:** 6
**Confidence:** 4

**Summary:**

This work introduces RealChat-1M, a large-scale dataset containing 1M human-LLM conversations.  The authors presented a comprehensive exploration of the dataset through four distinct use cases: the development of content moderation models, establishment of a robust safety benchmark, training of instruction-following models, and the creation of challenging benchmark questions, highlighting the multifaceted utility of the dataset.

**Strengths:**

- A large-scale real-world conversational dataset with diverse range of applications, including the development of content moderation models, the creation of a robust safety benchmark (jailbreak conversations), the training of instruction-following models, and the collection of challenging benchmark questions (Arena-Hard-200).
- This work offers a comparative analysis across all use cases, leveraging both open-source LLMs like Vicuna and proprietary models such as GPT-4. The experimental results reveals 1) large performance gaps between open and proprietary models, 2) safety concerns of all the models especially the models developed by open source community (e.g., Vicuna, Alpaca).

**Weaknesses:**

While the presented dataset significantly surpasses the scale of other datasets, I find myself questioning the unique value it brings. OpenAssistant, SharedGPT, Anthropic HH, and Chatbot Arena have already proven their worth in fine-tuning LLMs, assessing LLM safety and performance. What compelling reasons can be put forth to encourage researchers to opt for a subset of RealChat-1M for these very purposes?

**Questions:**

See Weakness

---

> ### Author Response · Authors · 2023-11-22
>
> We thank reviewer M7pJ for their feedback and for recognizing the scale of our dataset. To address the weakness pointed out by the reviewer, we wish to highlight the unique aspects of our dataset and emphasize its importance in terms of scale:
>
> 1. Our dataset is up-to-date and will be continually updated. We actively maintain our data collection website and plan to release more data dumps in the future. Over the past month, we received approximately 25K user queries per day for the latest models, including GPT-4 Turbo, Mistra, and Zephyr. We also intend to release more data on human preferences (votes).
> 2. Our dataset is highly diverse. Unlike the datasets mentioned, which typically include only one model or a very limited number of models, ours encompasses a much wider range of models and user distributions.
> 3. The scale of the dataset is crucial. In an era where there is a growing interest in scraping the entire internet to train LLMs, we believe that scale is one of the most critical metrics for a dataset. The advancements in machine learning have often been driven by scaling up datasets, as evidenced by projects like ImageNet and GPT-3/4.

---

### Official Review · Reviewer_kFoy · 2023-10-30

**Soundness:** 3 good
**Presentation:** 3 good
**Contribution:** 4 excellent
**Rating:** 8
**Confidence:** 4

**Summary:**

This paper introduces RealChat-1M, a dataset contains 1 million real-world conversations with 25 large language models collected from 210,000 unique IP addresses in the wild. The authors demonstrate the usefulness of the dataset on multiple use cases like content moderation models, safety benchmarks, or training instruction-following models.

**Strengths:**

- The dataset introduced by this paper is a large-scale dataset containing interactive logs of 210,000 unique IP addresses with 25 large language models, which is both extremely valuable and meaningful for future LLM development, given most datasets that were used to train LLMs are not publicly available.
- Part of the data that can jailbreak the safeguards of leading LLMs is repurposed by the authors to be a benchmark for safety and robustness study.
- The authors also curate a benchmark that contains challenging and high-quality user prompts for identifying the gap between open-sourced and proprietary models.
- Demonstration for the use cases of the dataset were nicely done.

**Weaknesses:**

- Figure 1 and 2 could be redrawn by leaving vicuna/English out because the distribution is left-skewed and therefore the number for the rest of the models/languages are hard to interpret.
- Although in the limitation section there is a paragraph about the data quality, deeper analysis could be done. For example, how many of them is from MMLU and MT-Bench, is there some human annotation done for duplicate data?

**Questions:**

How many prompts are there after filtering for section 3.2?

---

> ### Author Response · Authors · 2023-11-22
>
> We thank reviewer kFoy for their feedback.
>
> > Figure 1 and 2 could be redrawn by leaving vicuna/English out
>
> We will consider redrawing Figures 1 and 2, either by using a log scale or by splitting them into two separate figures.
>
> > Although in the limitation section there is a paragraph about the data quality, deeper analysis could be done. For example, how many of them is from MMLU and MT-Bench, is there some human annotation done for duplicate data?
>
> We ran a string match to find the overlap between RealChat-1M and MT-bench questions. We found that 12 out of 160 questions from MT-bench are present in RealChat-1M with an exact string match. When performing topic clustering, we also applied duplication checks based on string matching.
>
> > How many prompts are there after filtering for section 3.2?
>
> We began with 100K conversations, each averaging two turns, resulting in 200K prompts. After filtering, half of the prompts were removed, leaving us with 100K prompts.

---

### Official Review · Reviewer_hTkb · 2023-10-31

**Soundness:** 4 excellent
**Presentation:** 3 good
**Contribution:** 3 good
**Rating:** 8
**Confidence:** 4

**Summary:**

This work presents a dataset of 1 million real user chats with 25 popular LLMs. The main contribution of the paper is the data itself, which was collected over 5 months and includes a diverse array of user interactions. The authors present some analysis of the contents of the dataset, then show 4 use cases: building a moderation model, a safety benchmark, instruction tuning, and a challenging prompt benchmark.

**Strengths:**

- The data itself is quite valuable. As the authors note, much of the data actually used in training models is proprietary and private
- The 4 use cases are all creative and well designed. They demonstrate the potential of the dataset as a strong resource
- The analysis is quite interesting, for example the cluster analysis in figure 3. It also supports/is validated by past work, e.g. the observation that many users now are interested in LLMs for help with coding
- The size and diversity of the dataset promises many future uses

**Weaknesses:**

- The dataset does not seem quite as diverse as the abstract suggests. Although 25 LLMs are used, Vicuna-13B is by far the most frequently used one. Similarly, while there may be a few instances of many languages, the vast majority is still in English
- As the authors point out, there are no human preference values which is one weakness compared to related datasets (see table 1)
- It is not clear whether the use format (single model vs side-by-side) and IP address have similar balance issues to model and language above. Although there are 210K unique IP addresses, is some large portion of the dataset covered by only a few of these? And do most users access single models rather than the perhaps more interesting side-by-side mode? Having the answers to these questions would help evaluate this paper further.

**Questions:**

See weaknesses

**Details Of Ethics Concerns:**

It would be useful to ensure that the real user inputs do not put any of the users at risk of deanonymization or harm. The authors make a best effort to prevent this, but having some review of this may be necessary.

---

> ### Author Response · Authors · 2023-11-22
>
> Thank you for your questions and comments!
>
> > The dataset does not seem quite as diverse as the abstract suggests. Although 25 LLMs are used, Vicuna-13B is by far the most frequently used one. Similarly, while there may be a few instances of many languages, the vast majority is still in English
> > As the authors point out, there are no human preference values which is one weakness compared to related datasets (see table 1)
>
> It is true that Vicuna-13B and English comprise a large fraction of our dataset. This is because Vicuna-13B is the first model we hosted on the platform and the fact that most of our users speak English. Meanwhile, our dataset will continue to be updated. This is expected to mitigate the bias present in earlier models and keep pace with the latest data from new models. We are actively maintaining our data collection website and plan to release more data dumps in the future. In the month following our paper submission, we received approximately 25K user queries per day for the latest models, including GPT-4 Turbo, Mistra, and Zephyr. We also plan to release more data on human preferences, such as votes.
>
> > It is not clear whether the use format (single model vs side-by-side) and IP address have similar balance issues to model and language above. Although there are 210K unique IP addresses, is some large portion of the dataset covered by only a few of these? And do most users access single models rather than the perhaps more interesting side-by-side mode? Having the answers to these questions would help evaluate this paper further.
>
> The IP addresses are actually very sparse, so there is no small set of IP addresses that can cover a large portion of the dataset. It is true that most users access single models rather than side-by-side. But the distribution is shifting towards side-by-side more recently because we are promoting more on the side-by-side mode and making it the landing page.

---

> > ### Comment · Reviewer_hTkb · 2023-11-23
> >
> > Thank you for a detailed response, this answers my questions.

---

### Official Review · Reviewer_D9AT · 2023-11-01

**Soundness:** 3 good
**Presentation:** 3 good
**Contribution:** 4 excellent
**Rating:** 8
**Confidence:** 4

**Summary:**

Through the introduction of the RealChat-1M dataset, this paper provides a comprehensive resource for studying user interactions with LLMs in real-world settings. They leverage RealChat-1M for 4 use cases: developing content moderation models, building a safety benchmark, training instruction-following models, and creating challenging benchmark questions.

**Strengths:**

1. With a size of 1 million, this dataset stands out as unparalleled among its peers.
2. This dataset offers authentic user prompts and highlights potential safety concerns, paving the way for future research.
3. This paper presents studies that highlight the four distinct applications of this novel dataset.
4. The authors have a plan to regularly update the dataset in the future, which is a crucial step given the frequent release of newer and more advanced LLMs from the community.

**Weaknesses:**

While the user prompts in this dataset are genuine, the responses are synthetic and do not have quality ratings. Thus, additional processing is necessary before use.

**Questions:**

Are there plans to enhance the website's interface design to increase its accessibility for a broader audience? Such improvements could lead to a more diverse user base and a richer dataset distribution.

---

> ### Author Response · Authors · 2023-11-22
>
> We thank the reviewer for the feedback.
>
> > While the user prompts in this dataset are genuine, the responses are synthetic and do not have quality ratings. Thus, additional processing is necessary before use.
>
> Although limited, our data does include some user ratings (e.g., upvote/downvote) on model responses, in which the upvoted responses were used to train the model in Section 4.3. We will soon release the user feedback dataset after finishing the in-depth study and data cleaning.
>
> > Are there plans to enhance the website's interface design to increase its accessibility for a broader audience? Such improvements could lead to a more diverse user base and a richer dataset distribution.
>
> Yes, we’re actively enhancing the website’s interface and committed to growing this platform to a broader user base. In the past few weeks, our traffic has increased significantly due to active community engagement and now we’ve collected over five million user chat queries. Those new data will be included in our next dataset release. Please kindly let us know if you have any suggestions on UI/UX, thanks!

---

> > ### Comment · Reviewer_D9AT · 2023-11-23
> >
> > Thanks for the author's response, I have carefully read them.

---

### Meta-Review · Area_Chair_Ff71 · 2023-12-08

**Metareview:**

**Paper Summary:**

This paper introduces RealChat-1M, a dataset of one million real-world conversations between users and 25 large language models (LLMs). The authors show four applications of the dataset: developing content moderation models, constructing a safety benchmark, training instruction-following models, and creating a challenging benchmark.

**Strengths:**

1. Scale: The dataset's size of one million conversations collected from 210K unique IPs provide a comprehensive resource of authentic user prompts (D9AT, kFoy, M7pJ).
2. Utility: This paper shows the dataset's utility in four applications (D9AT, hTkb, kFoy, M7pJ)

**Weaknesses:**

1. Limited Quality and Diversity: Despite including 25 LLMs, the dataset is predominantly skewed towards Vicuna and English language conversations, which may limit the quality and diversity of the dataset (hTkb, kFoy).
2. Lack of Quality Annotations: The responses in the dataset come from LLMs but don't have quality ratings. (D9AT).

**Decision:**

Based on reviews, I recommend accepting this paper as a spotlight. The scale and real-world nature of the dataset, along with its potential applications, present a contribution to the field of LLMs. However, the authors should address the noted weaknesses, particularly in fixing Figures 1 and 2 to clearly visualize the tail due to the skewed distributions in models and languages.

**Justification For Why Not Higher Score:**

N/A

**Justification For Why Not Lower Score:**

I recommended spotlight since most reviews are very positive (8/8/8/6) and I think this dataset would be useful in future LLM research.

---

### Decision · Program_Chairs · 2024-01-16

Accept (spotlight)